# FIN−H₂AN: FINGERPRINT-BASED HETEROGENEOUS HYPERGRAPH ATTENTION NETWORK FOR MOLECULAR PROPERTY PREDICTION

## ABSTRACT

Molecular property prediction plays a crucial role in drug discovery and material design but faces major challenges such as constructing meaningful molecular features and defining effective similarity metrics between molecules. Recently, graph neural networks (GNNs) have shown great success in learning molecular representations from graphs via capturing local atomic interactions. However, they often struggle to capture global chemical motifs and complex long-range dependencies between molecules and substructures. In this paper, we present `Fin-H₂AN`, a novel Fingerprint-based Heterogeneous Hypergraph Attention Network to address the molecular property prediction problem. In this model, we propose a novel heterogeneous hypergraph structure by defining higher-order relations between molecules by leveraging diverse substructures derived from multiple molecular fingerprints. In this way, all molecules and their different fingerprints are embedded into a unified hypergraph. We employ a heterogeneous hypergraph attention model to learn meaningful molecular representations from the proposed heterogeneous hypergraph. It captures higher-order relations among molecules and integrates unique features of different molecular fingerprints into the molecular embeddings. These embeddings are then used for molecular property prediction. Extensive experiments on eight MoleculeNet benchmark datasets demonstrate that `Fin-H₂AN` outperforms the state-of-the-art molecular property prediction models and show its effectiveness in capturing both local and global molecular information.

## 1 INTRODUCTION

The rapid and efficient design of new drugs is essential for combating diseases, tackling pandemics and improving human health. Molecular property prediction is a crucial step in drug discovery and material design, as it enables the rapid identification of compounds with desired biological and physicochemical attributes. With the availability of public databases, several computational models have been proposed for molecular property prediction Zhang et al. (2024); Rollins et al. (2024); Nguyen-Vo et al. (2024). A major challenge in effectively utilizing machine learning models for this problem lies in constructing meaningful molecular representations and features, as well as defining effective similarity metrics between molecules Wigh et al. (2022); López-Pérez et al. (2024). Traditional molecular property prediction models, like quantitative structure–activity (property) relationship (QSAR/QSPR) Muratov et al. (2020); Toropov & Toropova (2020), use a large set of handcrafted features such as Extended-connectivity fingerprints (ECFPs) Rogers & Hahn (2010). However, this process is labor intensive, and selecting the right features to capture complex multi-scale interactions within molecules remains a significant challenge Lewis (2005).

Recently, graph neural networks (GNNs) have shown great success in learning molecular representations from molecular graphs Withnall et al. (2020); Feinberg et al. (2020); Xiong et al. (2019). Each molecule is represented as an undirected molecular graph, where nodes represent the set of atoms in the molecule, and edges represent the set of chemical bonds between atoms, capturing molecular connectivity and bonding relationships. This graph-based representation preserves critical atomic-level information essential for capturing molecular information (e.g., bond connectivity, stereochemistry, ring systems) Coley et al. (2017). However, GNNs applied to simple molecular

graphs often struggle to model complex higher-order interactions and long-range dependencies between substructures and within molecules Giraldo et al. (2023). They also lack prior knowledge of the domain including molecular fragments in molecules.

Alternatively, *molecular fingerprints* provide essential information for encoding molecular structures Mayr et al. (2018); Moriwaki et al. (2018). These fixed-length binary vectors indicate the presence or absence of specific chemical fragments (e.g., rings, chains, or functional groups) and provide complementary information that enhances molecular representations Cao et al. (2015); Yap (2011); Cereto-Massagué et al. (2015). However, simply applying models like SVM and regular neural networks to these vectors often fails to capture complex molecular interactions, such as higher-order dependencies within and between molecules Wu et al. (2018); Mayr et al. (2018). Moreover, selecting the most effective fingerprint representation remains a challenge. Some approaches, such as FP-GNN Cai et al. (2022), address this by concatenating multiple fingerprint features into a single vector, assuming an equal contribution from all bits while neglecting the relational context among chemical fragments. According to the Similar Property Principle, structurally similar molecules tend to exhibit similar properties and two molecules are similar if they have similar substructures as functional groups in their sequence Johnson et al. (1990); Lapez-Parez et al. (2024); Muegge & Mukherjee (2016). Finding similarities of molecules based on their substructures is a challenging task. Thus, there is a strong need for methods that better capture higher-order molecular interactions by explicitly considering the substructures encoded in fingerprints as domain knowledge.

To overcome these limitations, we develop a novel Fingerprint-based Heterogeneous Hypergraph Attention Network, `Fin-H₂AN`. To properly depict the structural similarity between molecules, we construct a novel heterogeneous hypergraph by using diverse substructures derived from multiple molecular fingerprints. In this hypergraph, we incorporate three different molecular fingerprints; MACCS, ErG, and PubChem Durant et al. (2002); Stiefl et al. (2006); Bolton et al. (2008). A hypergraph is a unique model of a graph with hyperedges. Unlike a regular graph where the degree of each edge is two, hyperedge as degree-free can connect an arbitrary number of nodes Bretto (2013). Each bit in a fingerprint, corresponding to a specific chemical fragment in the molecule, is represented as a node in the hypergraph. Each molecule is represented as a hyperedge connecting the nodes corresponding to its active fingerprint bits. The resulting hypergraph captures higher-order relations between molecules. Each hyperedge combines nodes derived from the three fingerprints, making it heterogeneous. To effectively encode this unified heterogeneous hypergraph and learn the representations of molecules, we employ a heterogeneous hypergraph attention model including dual attention mechanism. First, hyperedge-to-node attention aggregates the global molecular context into each fingerprint node by computing attention weights from all connected molecules. Second, node-to-hyperedge attention enables each molecule to aggregate refined fingerprint node representations, taking into account the heterogeneity of nodes. This results in a comprehensive, molecule-level embedding capturing higher-order relations among molecules and integrates unique features of different molecular fingerprints into the molecular embeddings. To evaluate the effectiveness of our model, we perform extensive experiments on eight MoleculeNet benchmark datasets and compare its performance with state-of-the-art baseline models for molecular property prediction. Our experimental results show that `Fin-H₂AN` achieves top-ranked performance, outperforming the state-of-the-art overall. Our main contributions are summarized as follows:

- We propose a unified fingerprint-based heterogenous hypergraph representations, where each molecule is a hyperedge connecting diverse fingerprint nodes (MACCS, ErG, and PubChem). This design preserves the unique signals of each fingerprint type and captures higher-order interactions among fragments and molecules.

- We design a heterogeneous hypergraph encoder with a dual attention mechanism: (i) hyperedge-to-node attention to inject global molecular context into fingerprint nodes, and (ii) node-to-hyperedge attention to aggregate refined node features into molecule-level embeddings. This setup highlights the importance of specific substructures for predicting molecular properties.

- We conduct extensive experiments on MoleculeNet benchmarks, showing that our model consistently outperforms state-of-the-art baselines.

## 2    RELATED WORK

**Single-View Models (Graphs and Fingerprints):**   Graph neural networks (GNNs) model
molecules as graphs where atoms are represented as nodes and bonds as edges.   Initial GNN ar-
chitectures, including graph convolutional networks (GCNs), graph attention networks (GATs) and
message passing neural networks (MPNNs), effectively captured local atomic relationships within
molecules Kipf & Welling (2016); Velickovic et al. (2017); Yang et al. (2019). Building on these
foundations, several advanced GNN models have been proposed. AttentiveFP adds attention to high-
light key local contexts Xiong et al. (2019); TrimNet fuses atom–bond–atom triplets with multi-head
attention to sharpen bond reasoning while limiting parameters Li et al. (2021); HiGNN decomposes
molecules with BRICS Degen et al. (2008) and applies feature-wise attention for multi-scale con-
text Zhu et al. (2022); ResGAT uses residual connections to stabilize training and extend effective
range Nguyen-Vo et al. (2024). Despite strong 2D topology modeling, atom-level message passing
can require many steps for distant motifs to interact, limiting long-range dependency capture for
properties.   As a result, important interactions that influence properties may be underrepresented.
Fingerprints encode chemically meaningful fragments into fixed-length vectors for efficient, inter-
pretable QSAR/QSPR Cherkasov et al. (2014); Mayr et al. (2018); Moriwaki et al. (2018). Fully-
connected neural networks (FCNNs) over ECFP or MACCS keys treat all fingerprint bits equally
ignoring task-specific importance and contextual relationships among fragments Cherkasov et al.
(2014).   Bypass architecture Wu et al. (2018) adds residual shortcuts between raw fingerprints
and learned embeddings to preserve information flow, but still relies on bit-wise independence.
FP2VEC Jeon & Kim (2019) learns embeddings for active bits and aggregates them to reveal pre-
dictive motifs.  These methods remain limited in explicitly modeling interactions or co-occurrence
patterns between chemically distant motifs, affecting their capacity to capture higher-order relation-
ships that affect the molecular property prediction.

**Fusion & Higher-Order Methods** To overcome the short-coming of single-view representations,
recent methods fuse multiple molecular modalities. FP-GNN Cai et al. (2022) concatenates finger-
prints embeddings, treating all fingerprint types equally, with atom-level GNN features, blending
local structure and global fragment cues. FP-BERT Wen et al. (2022) adapts masked-language pre-
training to ECFP substructures by treating bits as words and molecules as sentences, then fine-tunes
a CNN head for property prediction. MolPROP Rollins et al. (2024) fuses a pretrained SMILES-
BERT (ChemBERTa-2 Ahmad et al. (2022)) embeddings with GNN node features through joint
fine-tuning. PremuNet Zhang et al. (2024) advances this idea with two complementary pre-training
branches, one fusing SMILES, fingerprints and 2D graphs via a Transformer+GNN, the other mask-
ing and reconstructing 2D topology and 3D coordinates, to produce unified representations.  Al-
though these fusion strategies capture richer information, they can be computationally expensive,
and require large pre-training data. Hypergraphs generalize pairwise edges to hyperedges, directly
modeling multi-fragment co-occurrence Bretto (2013). Seq-HyGAN forms hyperedges over over-
lapping $k$-mers with attention Saifuddin et al. (2023), but arbitrary $k$-mer splits may miss chemically
meaningful units.  In contrast, we build a *heterogeneous* hypergraph whose nodes are well-defined
fingerprint bits (PubChem, MACCS, ErG) and whose hyperedges are molecules. A dual-attention
mechanism (type-aware and molecule-aware) enriches each bit with global molecular context and
then aggregates salient bits into a molecule embedding, preserving modality-specific semantics
while capturing long-range fragment interactions in a single step and avoiding heavy pretraining
or complex late fusion.

## 3    METHODOLOGY

In this section, we introduce our `Fin-H₂AN` model. We first describe how to build our fingerprint-
based heterogeneous hypergraph from three complementary fingerprint types (Section 3.1). Next,
we detail the dual attention heterogeneous hypergraph encoder, covering both hyperedge-to-node
and node-to-hyperedge message-passing stages(Section 3.2). Finally, we present the downstream
prediction head and loss functions (Section 3.3). Figure 1 gives a high-level view of our pipeline.

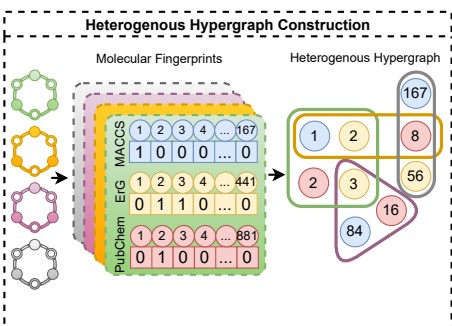 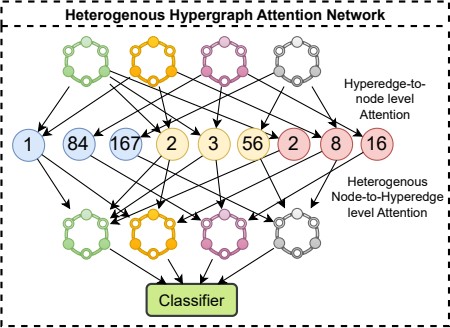

(a) Molecular Fingerprint based Heterogeneous Hypergraph Construction

(b) Molecular Fingerprint based Heterogeneous Hypergraph Attention Network

Figure 1: System architecture of the proposed method `Fin-H₂AN`. a) The first step of the model is heterogeneous hypergraph contraction, where each molecule is represented as hyperedge and bits fingerprints as substructures of molecules are represented as nodes. b) The second step is Heterogeneous Hypergraph Attention network with dual level attention as hyperedge-to-node level attention and node-to-hyperedge level attention

## 3.1 MOLECULAR FINGERPRINT-BASED HETEROGENEOUS HYPERGRAPH REPRESENTATION

One of the most widely used representations of molecular structures is SMILES (Simplified Molecular Input Line Entry System) Weininger (1988), that provides a compact and human-readable format to describe molecular structures. SMILES representations can be used to generate various molecular features, including molecular fingerprints and molecular graphs. Molecular fingerprints are designed to capture structural features of a molecule. There are different types of fingerprints and each type represents a group of specific intrinsic characteristics of a molecule, including structural motifs, connectivity, and functional groups. These fingerprints are represented as binary vectors, where each bit indicates the presence or absence of a specific substructural feature. To enhance molecular property prediction, we utilize three complementary fingerprint types: MACCS, ErG, and PubChem. Each of these fingerprints provides unique insights into molecular structures and enables a more comprehensive representation of molecular characteristics. MACCS fingerprints consist of 167 bits derived from predefined SMARTS (SMiles ARbitrary Target Specification) patterns, which represent common chemical substructures in organic compounds Durant et al. (2002). In contrast, ErG fingerprints (441 bits) employ a 2D pharmacophore-based approach, emphasizing the local spatial arrangement of atoms and their neighborhoods. This is particularly useful for capturing bioactive conformations in drug discovery Stiefl et al. (2006). Meanwhile, PubChem fingerprints, composed of 881 bits, provide a more comprehensive chemical description by encompassing a wide range of substructures through numerous SMARTS-based features Bolton et al. (2008).

According to the Similar Property Principle Johnson et al. (1990), structurally similar molecules tend to have similar properties, and two molecules are similar if they have similar substructures as functional groups in their sequence. Substructural patterns are effectively captured by molecular fingerprints. When two molecules share many of these patterns, reflected in their fingerprints through common active bits, they exhibit a higher degree of similarity. Building on these insights, to effectively capture higher-order substructural similarities of molecules, we construct novel heterogeneous hypergraph representations of molecules using molecular fingerprints. A hypergraph is a unique model of a graph with hyperedges. Unlike a regular graph where the degree of each edge is two, hyperedge is degree-free and can connect an arbitrary number of nodes. Each bit in a fingerprint, corresponding to a specific chemical fragment in the molecule, is represented as a node in this hypergraph. Each molecule is represented as a hyperedge connecting nodes of the molecule's active bits in each fingerprint. Since we incorporate three different fingerprints, the resulting hypergraph contains three distinct node types, making it heterogeneous.

Next, we formally define a heterogeneous hypergraph and introduce our proposed heterogeneous hypergraph model.

**Heterogeneous Hypergraph:** *A heterogeneous hypergraph is a tuple $\mathcal{HG} = (\mathcal{V}, \mathcal{E}, \mathcal{T})$ where $\mathcal{V} = \{v_1, ..v_i.., v_n\}$ is the set of nodes, $\mathcal{E} = \{e_1, ..e_j.., e_m\}$ is the set of hyperedges and $\mathcal{T} = \{t_1, ..., t_k\}$ is the set of node types. A hyperedge is a special type of edge that can connect any number of nodes, unlike traditional graph edges, which link only two nodes. Similar to the adjacency matrix of a simple graph, a hypergraph can be represented by an incidence matrix $H^{n \times m}$ where $H_{ij} = 1$ if $v_i \in e_j$, otherwise 0 with $n$ is the number of nodes and $m$ is the number of hyperedges.*

In our heterogeneous hypergraph $\mathcal{HG} = (\mathcal{V}, \mathcal{E}, \mathcal{T})$, we take the set of node types as $\mathcal{T} = \{\text{MACCS}, \text{ErG}, \text{PubChem}\}$. Each bit $i$ in a fingerprint type $t$ ($t \in \mathcal{T}$) is represented as a node $v_{i,t} \in V_t$, where $V_t$ denotes all nodes corresponding to the fingerprint type $t$. So we have 167 nodes for MACCS, 441 nodes for ERg and 881 nodes for PubChem fingerprints. The complete set of nodes in the hypergraph is given by $V = \bigcup_{t \in \mathcal{T}} V_t$. Each molecule $j$ is modeled as a hyperedge $e_j$, which connects the nodes corresponding to its active bits in each fingerprint type. As a result, every hyperedge contains three distinct types of nodes, each derived from a different fingerprint. Rather than merging all fingerprints into a single representation, maintaining separate fingerprint types as node types allows us to preserve the unique chemical signals of each type. The hypergraph is represented by an incidence matrix $H \in \{0,1\}^{|V| \times |E|}$, where $H_{i,j} = 1$ if the node $v_i$ (belonging to a specific type of fingerprint) is associated with the molecule (hyperedge) $e_j$, and $H_{i,j} = 0$ otherwise. For example, green molecule on Figure 1(a), has 1 active bit at position 1 for MACCS so include blude node 1, 2 active bit at position 2 and 3 fro ERg so include yellow nodes 2 and 3 and it include 1 active bit for PubChem at position 2 so include red node 2. This heterogeneous hypergraph structure explicitly encodes higher-order molecular relationships and captures complex interactions beyond pairwise similarities. Instead of representing every molecule as a graph, we represent all molecules in one hypergraph that capture functional similarities and this provides computational efficiency in addition to its effectiveness for molecular property prediction.

## 3.2 Heterogeneous Hypergraph Attention Network

In this section, we present our heterogeneous hypergraph attention network, which learns fingerprint-based molecular embeddings by explicitly modeling higher-order substructural similarities of molecules. Unlike standard GNN models that focus on node embeddings through pairwise relations, our model captures molecule-level higher order information through hyperedges. In our design, molecules are modeled as hyperedges that connect fingerprint nodes from three distinct types (MACCS, ErG, and PubChem). Each molecule is represented as a hyperedge linking the fingerprint nodes that correspond to its substructures. To capture this information, we use a dual attention mechanism that (i) enhances local fingerprint node embeddings with global molecular context through hyperedge-to-node attention, and (ii) aggregates the refined node features into a molecule-level representation using node-to-hyperedge attention, resulting in more comprehensive molecular embeddings for improved molecular property prediction.

### 3.2.1 Stage 1: Hyperedge-to-Node Attention

In this stage, each fingerprint node $v_{i,t}$ (node $i$ of fingerprint type $t \in \mathcal{T}$), represented by it's embedding $n_{i,t}$, aggregates information from all molecule hyperedges connected to it. While a fingerprint node represents a specific structural pattern, its importance is determined by the global context of the molecule. To model this relationship, we introduce an attention mechanism during the message-passing process from hyperedges to nodes. This mechanism learns the relevance of each molecule to the fingerprint node with respect to its type, enabling the model to dynamically weigh the contributions of different hyperedges based on their contextual importance. Using this attention mechanism, the updated representation of the fingerprint node is defined as:

$$n_{i,t}^{\text{new}} = \sum_{j \in \mathcal{E}(i)} \alpha_{j,i,t} \, W_1 \, e_j \tag{1}$$

where $W_1$ is a trainable weight and $\alpha_{j,i,t}$ is the attention coefficient that quantifies the importance of hyperedge $e_j$ to node $v_{i,t}$. The attention coefficient with respect to the node type is computed as:

$$\alpha_{j,i,t} = \frac{\exp(a_{i,j,t})}{\sum\limits_{j' \in \mathcal{E}(i)} \exp(a_{i,j',t})}, \quad a_{i,j,t} = \text{LeakyReLU}\big(W_2 \, e_j \cdot W_{3,t} \, n_{i,t}\big) \tag{2}$$

where $\mathcal{E}(i)$ is the set of molecules connected to node $i$ and $W_2$ and $W_{3,t}$ are trainable weights. $W_{3,t}$ is type-specific to adapt to the unique characteristics of each fingerprint type.

### 3.2.2 STAGE 2: NODE-TO-HYPEREDGE ATTENTION

The second stage aggregates enhanced node embeddings into comprehensive molecule-level embeddings. The node-to-hyperedge attention layer captures the most relevant structural patterns from each fingerprint type (MACCS, ErG and PubChem) and combines them into a comprehensive molecular representation. The aggregated hyperedge embedding $z_{j,HyG}$ is:

$$z_j = MLP([e_{j,\text{MACCS}}^{\text{new}} \,\|\, e_{j,\text{ErG}}^{\text{new}} \,\|\, e_{j,\text{PubChem}}^{\text{new}}]) \tag{3}$$

where $e_{j,\text{t}}^{\text{new}}$, for fingerprint type $t$, represents the updated representation of molecule $j$ specifically aggregated from fingerprint nodes of type $t$. It is computed as:

$$e_{j,t}^{\text{new}} = \sum_{i \in \mathcal{N}(j,t)} \alpha_{i,t,j} \, W_{6,t} n_{i,t}^{\text{new}} \tag{4}$$

The attention coefficient $\alpha_{i,t,j}$, representing fingerprint node $v_{i,t}$ contribution to molecule $e_j$, is defined as:

$$\alpha_{i,t,j} = \frac{\exp(a_{j,i,t})}{\sum\limits_{i' \in \mathcal{N}(j,t)} \exp(a_{j,i',t}))}, \quad a_{i,t,j} = \text{LeakyReLU}\big(W_{4,t} n_{i,t}^{\text{new}} \cdot W_5 e_{\text{j}}\big)\big) \tag{5}$$

where $\mathcal{N}(j,t)$ is the set of nodes of type $t$ connected to hyperedge $j$, and $W_{4,t}$, $W_5$ and $W_{6,t}$ are learnable weight matrices, with $W_{4,t}$ and $W_{6,t}$ being type specific.

### 3.3 PREDICTION AND OPTIMIZATION

After learning the representations of molecules, our goal is to predict molecular properties using these representations. For this, we utilize a Multilayer perceptron (MLP) that takes the representation $z_j$ of a molecule $j$ and generates the final output as whether molecule $j$ have a specific property

$$\hat{y}_j = \text{MLP}(z_j) \tag{6}$$

The entire model is trained using a binary cross-entropy (BCE) loss function for classification tasks and a mean squared error (MSE) loss function for regression tasks, and are computed as follows:

$$\mathcal{L}_{\text{BCE}} = -\frac{1}{N} \sum_{j=1}^{N} [y_j \log(\hat{y}_j) + (1 - y_j) \log(1 - \hat{y}_j)] \tag{7}$$

$$\mathcal{L}_{\text{MSE}} = \frac{1}{N} \sum_{j=1}^{N} (y_j - \hat{y}_j)^2 \tag{8}$$

where $N$ is the total number of samples, $y_j$ is the ground truth label (or value) and $\hat{y}_j$ is the predicted label (or value).

## 4 EXPERIMENTS

### 4.1 DATASETS

We evaluate our proposed model on eight widely-used benchmark datasets from MoleculeNet Wu et al. (2018). For classification tasks, we use datasets that evaluate different aspects of drug bioactivity and toxicity. Specifically, the BACE dataset is employed to predict $\beta$-secretase inhibition, while BBBP focuses on blood-brain barrier permeability. The ClinTox dataset is used to distinguish between FDA-approved compounds and those that failed due to safety concerns, and Tox21 consists of 12 independent toxicity assays. Additionally, the SIDER dataset examines adverse drug reactions by categorizing 27 different side effects. For regression tasks that predict key physicochemical properties of molecules, we use three datasets: the ESOL dataset for aqueous solubility, Lipophilicity for the octanol-water partition coefficient, and FreeSolv for hydration free energy Wu et al. (2018). More information about the datasets can be found in Appendix (Table 6).

## 4.2 EXPERIMENTAL SETUP

For each dataset, we adopt a random split strategy, allocating 80% of the data for training, 10% for validation, and 10% for testing. To mitigate sampling bias, we repeat each experiment over 10 different random seeds and report the averaged metrics. Models are trained for up to 100 epochs using the Adam optimizer, with early stopping triggered if validation performance does not improve over 7 consecutive epochs. For classification tasks, we evaluate performance using ROC-AUC, averaging results when multiple labels are involved. For regression tasks, we report the Root Mean Squared Error (RMSE), where lower values signify better performance. A dynamic learning rate schedule is employed via a Noam scheduler, adjusting the rate from an initial $1 \times 10^{-5}$ up to a peak of $1 \times 10^{-3}$, and then decaying to a final value in the range of $5 \times 10^{-5}$ to $1 \times 10^{-4}$. Dropout rates are tuned per dataset; set to 0.10 for Lipophilicity, FreeSolv, and ESOL, 0.30 for BACE, BBBP, and Tox21, and increased to 0.50 for ClinTox and SIDER.

## 4.3 BASELINES

To assess Fin-H$_2$AN performance, we include three categories of baselines. First, foundational GNN architectures, GCN Kipf & Welling (2016), GAT Velickovic et al. (2017), and D-MPNN Yang et al. (2019), which provide convolutional, attention-based, and message-passing perspectives. Second, we include 3 models from advanced molecular graph models, which are AttentiveFP Xiong et al. (2019) that applies self-attention during message passing; TrimNet Li et al. (2021) that uses multi-head attention over atom–bond–atom triplets; and ResGAT Nguyen-Vo et al. (2024) that integrates residual connections into GAT layers. Third, as similar to our modality, two fingerprint-based models are included, which are FP-GNN Cai et al. (2022) that fuses learned fingerprint embeddings with graph features, and FP2VEC Jeon & Kim (2019) that learns dense bit embeddings. All baselines use the same data splits and evaluation protocols. For all baselines except FP-GNN and FP2Vec , we adopt published results from ResGAT Nguyen-Vo et al. (2024) to maintain consistency. For FP-GNN and FP2Vec methods, we retrain them using their recommended hyperparameters to ensure a fair comparison.

## 4.4 RESULTS AND ANALYSIS

Tables 1, 2, and 3 summarize the performance of our proposed Fin-H$_2$AN model alongside baselines across a diverse set of molecular property prediction tasks. Our evaluation covers five classification datasets (BACE, BBBP, ClinTox, SIDER, and Tox21) and three regression datasets (ESOL, Lipophilicity, FreeSolv). For classification, we report the ROC-AUC (with higher values indicating better performance), whereas for regression tasks, we present RMSE (with lower values indicating better performance). To provide a robust comparison, we employ a ranking-based evaluation: for each dataset, models are ordered based on their performance (with a rank of [1] assigned to the best result), and an average rank (Avg Rank) is computed across datasets. All results are averaged over 10 random splits.

**Classification Performance:** Table 1 summarizes ROC-AUC results with standard deviations and ranking per-dataset (1=best) for tree classification dataset. Fin-H$_2$AN achieves first-place performance on BACE (88.77 $\pm$ 2, rank 1) and BBBP (92.84±1, rank 1), improving over the second-best FP-GNN by 0.30–0.94 points while reducing the standard deviation on BBBP from 3 to 1. On SIDER, Fin-H$_2$AN scores 65.00 $\pm$ 2 (rank 2), narrowly trailing FP2VEC's 65.73 $\pm$ 1; both methods markedly outperform the rest, but our model still cuts baseline variability in half. In terms of average rank across all datasets, Fin-H$_2$ANachieves the top position with a score of 1.3, indicating consistent superiority over both classical GNN baselines (GCN, GAT) and recent state-of-the-art models such as D-MPNN, AttentiveFP, and FP-GNN. Table 2 show the performance on toxicity classification datasets Tox21 and ClinTox. Our model and other fingerprint-based models are outer performed by several deep GNN baselines. In ClinTox, ResGAT achieves 88.81 $\pm$ 1 (rank 1) and D-MPNN 85.31 $\pm$ 1 (rank 2) versus our 79.58 $\pm$ 6 (rank 4), while on Tox21 AttentiveFP reaches 86.00 $\pm$ 5 (rank 1) and D-MPNN 84.79 $\pm$ 5 (rank 2) versus our 84.34 $\pm$ 1 (rank 4), although Fin-H$_2$AN still outperforms other fingerprint-based models. This underperformance likely stems from the complex, multi-target toxicity endpoints in these datasets, where atom-level stereoelectronic features, better captured by deep GNNs, play a dominant role Gilmer et al. (2017). Across all five tasks, Fin-H$_2$AN achieves an average classification rank of 2.4, the best among all compared

methods, demonstrating that its heterogeneous hypergraph attention yields both high mean accuracy and low variance. On the other hand, data modality is important for different datasets. While toxicity datasets detailed atom level information is more important, for other datasets, substructures plays more important roles for molecular property prediction.

Table 1: ROC-AUC Results on MoleculeNet classification datasets.

| Model | BACE | BBBP | SIDER | Avg Rank |
|---|---|---|---|---|
| GCN* | $65.05 \pm 3^8$ | $61.55 \pm 6^8$ | $53.92 \pm 4^8$ | 8 |
| GAT* | $63.16 \pm 6^9$ | $63.38 \pm 7^9$ | $56.18 \pm 5^9$ | 9 |
| D-MPNN* | $87.88 \pm 4^5$ | $91.97 \pm 3^2$ | $59.21 \pm 5^6$ | 4.3 |
| AttentiveFP* | $88.36 \pm 4^4$ | $91.29 \pm 4^4$ | $64.09 \pm 6^4$ | 4 |
| TrimNet* | $80.54 \pm 5^7$ | $82.75 \pm 3^7$ | $54.56 \pm 7^7$ | 7 |
| ResGAT* | $88.40 \pm 3^3$ | $90.77 \pm 2^6$ | $63.00 \pm 5^5$ | 4.6 |
| FP-GNN | $88.47 \pm 2^2$ | $91.90 \pm 1^3$ | $64.97 \pm 1^3$ | 2.6 |
| FP2VEC | $87.81 \pm 2^6$ | $90.88 \pm 3^5$ | $65.73 \pm 1^1$ | 4 |
| **Fin–H$_2$AN (Ours)** | $\mathbf{88.77 \pm 2^1}$ | $\mathbf{92.84 \pm 1^1}$ | $65.00 \pm 2^2$ | **1.3** |

Table 2: ROC-AUC Results on MoleculeNet Toxicity classification datasets.

| Model | ClinTox | Tox21 | Avg Rank |
|---|---|---|---|
| GCN* | $46.68 \pm 3^9$ | $60.78 \pm 4^8$ | 8.5 |
| GAT* | $53.29 \pm 7^8$ | $48.28 \pm 4^9$ | 8.5 |
| D-MPNN* | $85.31 \pm 1^2$ | $84.79 \pm 5^2$ | 2 |
| AttentiveFP* | $84.59 \pm 2^3$ | $86.00 \pm 5^1$ | 2 |
| TrimNet* | $69.70 \pm 6^7$ | $84.51 \pm 4^3$ | 5 |
| ResGAT* | $88.81 \pm 1^1$ | $83.97 \pm 5^6$ | 3.5 |
| FP-GNN | $78.46 \pm 8^5$ | $84.05 \pm 1^5$ | 5 |
| FP2VEC | $76.02 \pm 6^6$ | $79.70 \pm 2^7$ | 6.5 |
| **Fin–H$_2$AN (Ours)** | $79.58 \pm 6^4$ | $84.34 \pm 1^4$ | **4** |

**Regression Performance:** Table 3 reports RMSE results, with standard deviations and ranking of models for each dataset from ESOL, Lipophilicity, and FreeSolv for regression tasks. These datasets evaluate a model's ability to predict key molecular properties such as solubility and hydration free energy, which are critical in drug discovery and molecular design. Our model achieves the best performance with lowest error on ESOL ($0.6487 \pm 0.04$, rank 1) and FreeSolv ($1.0115 \pm 0.12$, rank 1), outperforming strong message-passing baselines such as D-MPNN by 6–10% while maintaining the standard deviation among the smallest. On Lipophilicity, Fin–H$_2$AN ranks second ($0.6254 \pm 0.03$, rank 2), narrowly behind D-MPNN ($0.6148 \pm 0.03$) but again with identical low variance. These results yield an average regression rank of 1.3 for our model, demonstrating consistent and superior performance compared to both classical GNN baselines (e.g., GCN, GAT) and recent state-of-the-art models (e.g., D-MPNN, FP-GNN). For visual analysis, we provided predicted versus ground truth plots in the Appendix (Figures 2 (a, b, c)), which illustrates that the predictions closely align with the diagonal $\hat{y} = y$, supporting the low RMSE and variance reported.

Table 3: ROC-AUC Results on MoleculeNet regression datasets.

| Model | ESOL | Lipophilicity | FreeSolv | Avg Rank |
|---|---|---|---|---|
| GCN* | $2.0569 \pm 0.14^8$ | $1.1974 \pm 0.03^7$ | $3.6618 \pm 0.75^8$ | 7.7 |
| GAT* | $2.4261 \pm 0.16^9$ | $1.4974 \pm 0.06^8$ | $4.4315 \pm 0.87^9$ | 8.7 |
| D-MPNN* | $0.6930 \pm 0.08^2$ | $\mathbf{0.6148 \pm 0.03^1}$ | $1.1394 \pm 0.18^3$ | 2.0 |
| AttentiveFP* | $1.5225 \pm 0.13^7$ | $1.1232 \pm 0.03^6$ | $3.5585 \pm 0.72^7$ | 6.7 |
| TrimNet* | $0.7499 \pm 0.04^4$ | $0.6315 \pm 0.03^5$ | $1.5996 \pm 0.16^6$ | 5.0 |
| ResGAT* | $0.8125 \pm 0.07^5$ | $0.6833 \pm 0.02^4$ | $1.4734 \pm 0.31^5$ | 4.7 |
| FP-GNN | $0.7453 \pm 0.07^3$ | $0.6674 \pm 0.02^3$ | $1.0720 \pm 0.18^2$ | 2.7 |
| FP2VEC | $1.3134 \pm 0.14^6$ | $1.0055 \pm 0.04^9$ | $2.2506 \pm 0.54^4$ | 6.3 |
| **Fin–H$_2$AN (Ours)** | $\mathbf{0.6487 \pm 0.04^1}$ | $0.6254 \pm 0.03^2$ | $\mathbf{1.0115 \pm 0.12^1}$ | **1.3** |

## 4.5 Fingerprint Ablation Study

To quantify the individual contribution of each fingerprint modality, we compare four variants of our HFP-GIN model: (1) MACCS-only, (2) ErG-only, (3) PubChem-only, and (4) the full combination of all three fingerprint types. Tables 4 and 5 report the mean±std over 10 independent runs for classification (ROC-AUC) and regression (RMSE) benchmarks, respectively. While each fingerprint provides reasonable predictive power on its own, combining all three consistently yields the best performance across all datasets. On the classification tasks, the combined (MACCS+ErG+PubChem) model, `Fin-H₂AN`, outperforms each single-fingerprint variant by almost 1—16% percentage points in ROC-AUC. For example, on BACE the combined model achieves 88.77±2 whereas it achieves 68.66±5 for MACCS-only and the best single-fingerprint result is 81.59 (ErG) . On the regression tasks, the fusion model reduces RMSE by almost 16–36% compared to single-fingerprint baselines. Notably, on FreeSolv, ErG-only fails (no prediction) because fewer than 50% of molecules present ErG pharmacophores, forcing the combined model to rely on MACCS and PubChem features. These findings highlight the complementary nature of the fingerprints and demonstrate that their integration in `Fin-H₂AN` leads to significant performance gains in molecular classification tasks.

Table 4: Ablation study on Classification Datasets.

| Model | BACE | BBBP | SIDER | ClinTox | Tox21 |
|---|---|---|---|---|---|
| **MACCS only** | $68.66 \pm 5$ | $90.80 \pm 2$ | $55.18 \pm 3$ | $77.91 \pm 8$ | $77.70 \pm 2$ |
| **ErG only** | $81.59 \pm 4$ | $89.36 \pm 3$ | $56.25 \pm 3$ | $75.31 \pm 6$ | $69.37 \pm 1$ |
| **PubChem only** | $73.15 \pm 3$ | $91.79 \pm 1$ | $54.89 \pm 3$ | $73.17 \pm 6$ | $80.35 \pm 1$ |
| **MACCS+ErG+PubChem** | $\mathbf{88.77 \pm 2}$ | $\mathbf{92.84 \pm 1}$ | $\mathbf{65.00 \pm 2}$ | $\mathbf{79.58 \pm 6}$ | $\mathbf{84.34 \pm 1}$ |

Table 5: Ablation study on Regression Datasets.

| Model | ESOL | Lipophilicity | FreeSolv |
|---|---|---|---|
| **Maccs only** | $1.0487 \pm 0.06$ | $1.2285 \pm 0.05$ | $1.4558 \pm 0.19$ |
| **Erg only** | $1.7157 \pm 0.12$ | $0.9773 \pm 0.14$ | – |
| **Pubchem only** | $0.7788 \pm 0.03$ | $1.2284 \pm 0.05$ | $1.7599 \pm 0.79$ |
| **Maccs+ErG+Pubchem** | $\mathbf{0.6487 \pm 0.04}$ | $\mathbf{0.6254 \pm 0.03}$ | $\mathbf{1.0115 \pm 0.12}$ |

## 4.6 Case Study: Interpreting BBB Permeability via Fingerprint Contributions

To further interpret our model's predictions, we conducted a case study in blood-brain barrier (BBB) permeability using the BBBP dataset. We analyze fingerprint attributions and fragment-level ClogP values for two representative compounds, one BBB-permeable and one non-permeable. The detailed case study is provided in Appendix A.3.

## 5 Conclusion

In this work, we introduced `Fin-H₂AN`, a novel fingerprint-based heterogeneous hypergraph attention network that unifies multiple molecular fingerprints into a single hypergraph and uses a dual attention mechanism to capture both global molecular context and modality-specific substructure importance. By treating MACCS, ErG, and PubChem bits as distinct node types and molecules as hyperedges, `Fin-H₂AN` naturally models higher-order interactions among chemical fragments and rapidly aggregates long-range dependencies in a single message-passing step. Extensive evaluations on eight MoleculeNet benchmarks demonstrate that `Fin-H₂AN` consistently outperforms state-of-the-art graph-based and fingerprint-based models across both classification and regression tasks, while exhibiting low variance across random splits. Looking forward, we plan to enrich the hypergraph by adding atom-level nodes alongside multi-omics node types, such as gene expression, so a single dual-attention encoder can seamlessly integrate chemical, structural, and biological relationships without any separate fusion steps.

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

## A  APPENDIX

Code is available at: https://github.com/anonymous-researcher28/code/

### A.1  DATASETS

### A.2  REGRESSION PERFORMANCE PLOTS

Figures 2 (a, b, c) illustrate the predicted ($\hat{y}$) versus ground truth ($y$) values for Lipophilicity, ESOL, and FreeSolv, respectively. In all three datasets, the points closely follow the red diagonal line $\hat{y} = y$, indicating strong agreement between predictions and true values and underscoring the low RMSE and small standard deviations reported in Table 3.

Table 6: Dataset Summary

| Dataset | Task Type | #Tasks | #Compounds |
|---|---|---|---|
| BACE | Binary Classification | 1 | 1,513 |
| BBBP | Binary Classification | 1 | 2,039 |
| ClinTox | Multi-label Classification | 2 | 1,491 |
| Tox21 | Multi-label Classification | 12 | 7,831 |
| SIDER | Multi-label Classification | 27 | 1,427 |
| ESOL | Regression | 1 | 1,128 |
| Lipophilicity | Regression | 1 | 4,200 |
| FreeSolv | Regression | 1 | 642 |

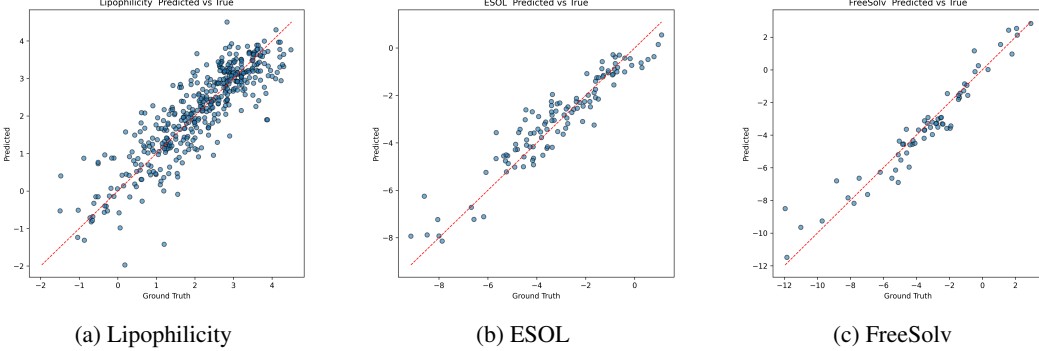

(a) Lipophilicity      (b) ESOL      (c) FreeSolv

Figure 2: Predicted vs. true scatter plots for regression datasets. Blue markers are individual molecule predictions; the red dashed line denotes perfect agreement ($\hat{y} = y$). The tight clustering around the diagonal highlights the low RMSE values reported in Table 3.

### A.3 CASE STUDY: INTERPRETING BBB PERMEABILITY VIA FINGERPRINT CONTRIBUTIONS

To evaluate the interpretability of our model, we analyze two representative compounds from the BBBP dataset. Molecule 1 (Fig. 3-a) is known to cross the blood–brain barrier (BBB), whereas Molecule 2 (Fig. 3-b) is not. Table 7 and 8 lists the top three bits per fingerprint and their corresponding substructure meanings for molecule 1 and molecule 2, respectively. We further computed fragment-level ClogP values with RDkit rdk (2024) to connect our model's attributions with chemical permeability features. ClogP, the logarithm of the octanol–water partition coefficient ($P$), is widely used to assess lipophilicity, an important factor for passive membrane diffusion Ghose et al. (1998); Wu et al. (2023). Molecules with a ClogP value around 2.5 tend to exhibit optimal BBB permeability, whereas low or negative ClogP values typically indicate poor lipid solubility and weak diffusion Pajouhesh & Lenz (2005).

In Molecule 1, the highest weight attributions point to the aromatic/halogenated fragment (MACCS 162: aromatic atom; PubChem 384: conjugated C=C; ErG 203: donor–acceptor/halogen contact). which is exactly the red-shaded phenyl ring in Fig. 3-a. This fragment shows a positive fragment ClogP consistent with a lipophilic patch that helps transmembrane diffusion Banks (2009). At the same time, the model also attends to featyres around amine/tropane region (MACCS 49: positively charged atom; PubChem 352/443: C–O and carbonyl; ErG 124/81: donor motifs) which aligns with the gray polar fragment showing a much lower fragment ClogP. The hydrophobic ring balanced by the polar cationic center is a classic pattern for BBB-permeable drugs where a sufficiently lipophilic fragment compensate local polarity to enable diffusion while maintaining recognition for solubility Pajouhesh & Lenz (2005); Wu et al. (2023). For the non-permeable compound, the model's top bits emphasize oxygenated donor rich fragments. These maps to the sugar-like scaffold highlighted in red in Fig. 3-b and neighboring polyol/amine regions, all showing negative fragment ClogP. This is the expected signature if high polarity and multiple H-bond donors/acceptors which reduce effective partitioning into lipid membranes Banks (2009); Pajouhesh & Lenz (2005).

Table 7: Top 3 Active Bits per Fingerprint Type for Molecule 1

| Fingerprint | Top Bits | Meaning |
|---|---|---|
| MACCS | 49 | positively charged atom |
| MACCS | 162 | aromatic atom |
| MACCS | 153 | heteroatom adjacent to $CH_2$ |
| PubChem | 384 | conjugated C=C bonds |
| PubChem | 352 | C–O single bond |
| PubChem | 443 | carbonyl group |
| ErG | 124 | ('Donor', 'Aromatic', 8 ) |
| ErG | 203 | ('DA', 'Halogen', 6 ) |
| ErG | 81 | ('Donor','Donor', 1 ) |

Table 8: Top 3 Active Bits per Fingerprint Type for Molecule 2

| Fingerprint | Top Bits | Meaning |
|---|---|---|
| MACCS | 137 | ('non-aromatic heterocycle atom') |
| MACCS | 153 | ('heteroatom adjacent to $CH_2$') |
| MACCS | 157 | ('C–O single bond') |
| PubChem | 352 | ('C–O single bond') |
| PubChem | 346 | ('aliphatic carbon with OH substituent') |
| PubChem | 284 | ('C–C single bond') |
| ErG | 124 | ('Donor', 'Aromatic', 8 ) |
| ErG | 81 | ('Donor','Donor', 1 ) |
| ErG | 151 | ('Donor', 'Ring', 8 ) |

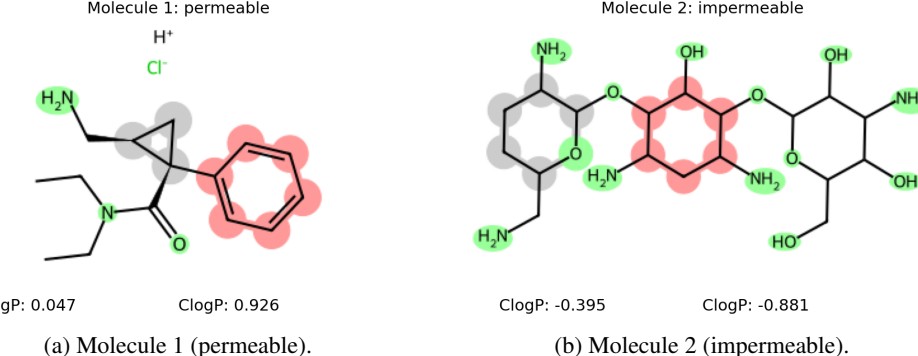

(a) Molecule 1 (permeable).      (b) Molecule 2 (impermeable).

Figure 3: Molecular representation and fragment ClogP values for a a) permeable b) impermeable molecules

