# OpenReview forum: "Fin-H2AN: Fingerprint-based Heterogeneous Hypergraph Attention Network for Molecular Property Prediction"
_ICLR.cc/2026/Conference — ICLR 2026 Conference Withdrawn Submission_

### Official Review · Reviewer_4LXH · 2025-10-23

**Soundness:** 3
**Presentation:** 3
**Contribution:** 2
**Rating:** 2
**Confidence:** 5

**Summary:**

This paper proposes Fin-H2AN, a fingerprint-based heterogeneous hypergraph attention network for molecular property prediction. The key idea is to represent molecules as hyperedges and fingerprint bits as nodes in a heterogeneous hypergraph, integrating three types of molecular fingerprints (MACCS, ErG, PubChem).

**Strengths:**

The paper introduces a heterogeneous hypergraph formulation of molecular data, which is relatively less explored compared to conventional GNN approaches.
The proposed dual attention mechanism provides a straightforward way to capture interactions between molecules and their substructures.
Empirical results on standard MoleculeNet datasets demonstrate some improvements over several classical baselines, and the ablation study illustrates the complementary effects of different fingerprints.

**Weaknesses:**

1. Unclear Motivation and Insufficient Problem Definition: The paper claims that existing GNN models fail to capture domain prior knowledge such as molecular fragments. However, many recent works already integrate molecular fragments or functional groups into graph-based or fingerprint-enhanced models. The novelty of this motivation is therefore limited and insufficiently articulated.
2.  Methodological Innovation is Limited: Using molecular fingerprints as nodes may not overcome known limitations: different molecules can share the same fingerprint vectors, leading to indistinguishability and limiting the model’s expressive power. This fundamental drawback is not discussed or mitigated.
3. Experimental Design is Relatively Simple: The evaluation benchmarks are standard but limited, and the baselines used are mostly classical models (e.g., GCN, GAT, D-MPNN, AttentiveFP). Recent and stronger baselines, such as CMPNN, GROVER, Uni-Mol are missing.
4. Literature Review is Outdated: Several cited works are relatively old (pre-2022), and recent progress in molecular representation learning, fragment-based GNNs is not covered. This weakens the paper’s positioning and novelty claim.

**Questions:**

1.	The paper argues that existing methods lack molecular fragment-level prior knowledge. However, many recent models [1-3] explicitly incorporate such knowledge. Could the authors clarify what is fundamentally different here and why prior works are insufficient?

2.	The use of molecular fingerprints as nodes is questionable, as different molecules can have identical fingerprints — a known limitation of fingerprint representations. How does the proposed approach address this potential loss of uniqueness?

3.	The experiments use a relatively small set of datasets and mostly older baselines. Could the authors evaluate their model on more diverse benchmarks or against more recent state-of-the-art models to better validate its effectiveness?

[1] Jiang, T., Yao, Q., Wang, Z., Bao, X., Yu, S., & Xuan, Q. (2025). Expert-Guided Substructure Information Bottleneck for Molecular Property Prediction. Journal of Chemical Information and Modeling, 65(15), 7887-7900.

[2] Zhang, Z., Liu, Q., Wang, H., Lu, C., & Lee, C. K. (2021). Motif-based graph self-supervised learning for molecular property prediction. Advances in Neural Information Processing Systems, 34, 15870-15882.

[3] Wang, R., Dai, H., Yang, C., Song, L., & Shi, C. (2024, August). Advancing molecule invariant representation via privileged substructure identification. In Proceedings of the 30th ACM SIGKDD Conference on Knowledge Discovery and Data Mining (pp. 3188-3199).

---

### Official Review · Reviewer_hXhF · 2025-10-31

**Soundness:** 2
**Presentation:** 2
**Contribution:** 2
**Rating:** 4
**Confidence:** 3

**Summary:**

The paper presents Fin-H₂AN, a novel deep learning model for molecular property prediction. It overcomes the limitations of GNNs, which struggle with long-range dependencies, and molecular fingerprints, which fail to model complex interactions. Fin-H₂AN uses a heterogeneous hypergraph representation, where nodes represent fingerprint bits and hyperedges represent molecules, capturing higher-order relationships. A dual-attention mechanism refines fragment node embeddings and aggregates them into molecule-level representations. Experiments on MoleculeNet benchmarks show that Fin-H₂AN outperforms state-of-the-art models, especially in predicting physicochemical properties.

**Strengths:**

1. The representation of molecules as hyperedges and fingerprint bits as nodes constitutes a novel abstraction. This 'fragment-centric' perspective departs from the mainstream 'atom-centric' (GNN) paradigm. The use of different FP types as distinct node types, which renders the hypergraph heterogeneous, is also an interesting design choice.

2. The paper is clearly organized and supported by sufficient experimental validation.

**Weaknesses:**

1. Appendix section A.1 DATASETS is empty?

2. The core of this method relies on the Similar Property Principle, modeling molecular similarity through the co-occurrence of molecular fingerprints (fragment patterns). However, fingerprints are fundamentally an abstract compression of molecular structure, resulting in information loss and ambiguity. This reliance on macro-fragment-pattern similarity may be insufficient to capture the subtle microscopic structural details that determine certain critical properties.

3. While the proposed method is validated using publicly available datasets, these datasets are inherently restricted to molecules belonging to specific, limited chemotypes or classes. Since real-world applications involve a much broader and more diverse chemical space, the practical utility and generalizability of the proposed method remain uncertain.

**Questions:**

1. How would Fin-H₂AN perform if it were deployed in a real-world scenario with properties that are unknown and classes that are highly diverse?

---

### Official Review · Reviewer_7rXo · 2025-10-31

**Soundness:** 2
**Presentation:** 3
**Contribution:** 2
**Rating:** 2
**Confidence:** 4

**Summary:**

This paper proposes Fin-H2AN, a model that constructs a heterogeneous hypergraph using three types of molecular fingerprints (MACCS, ErG, and PubChem). Molecules are represented as hyperedges connecting fingerprint bit nodes, and a dual-attention mechanism is employed to learn molecular embeddings for property prediction. Although the authors claim state-of-the-art performance on several MoleculeNet benchmarks, the contributions are limited in terms of methodological novelty, technical depth, and empirical validation.

**Strengths:**

The manuscript is well-structured and clearly written.
The paper introduces a model named Fin-H2AN for molecular property prediction.
The ablation study on the contributions of different fingerprints provides useful insights into their relative importance.

**Weaknesses:**

1. Insufficient evaluation datasets: To the best of my knowledge, MoleculeNet includes more datasets than those evaluated in this work. The authors should justify their selection of only five classification datasets. Moreover, the largest dataset used contains only around 7,000 molecules. Evaluation on larger-scale datasets (e.g., HIV, MUV) is necessary to validate the method's effectiveness and scalability.
2. Data splitting strategy: To better assess the model's generalization capability, the authors should include evaluations under scaffold splitting, which is more challenging and reflective of real-world scenarios.
3. Inadequate baselines: The comparison lacks several state-of-the-art baselines, such as Mole-BERT [1], Uni-Mol [2], and VideoMol [3]. Including these would strengthen the validity of the claimed advancements.
4. Limited fingerprint selection: With over 20 molecular fingerprints available, the authors should evaluate the model with additional fingerprints or combinations to better demonstrate the impact of fingerprint choice on performance.
5. Generalization concern: During inference, unseen molecules may not be accurately initialized as hyperedges, potentially limiting the model's generalization to new data.
6. Lack of computational efficiency analysis: Although the hypergraph structure is claimed to be computationally efficient, no comparison of training or inference time with other models is provided.
7. Underperformance on key datasets: The model performs poorly on SIDER, ClinTox, and Tox21 compared to GNN baselines like ResGAT and D-MPNN. The authors attribute this to the reliance on atomic-level stereoelectronic features but fail to provide further analysis or strategies for improvement.

[1] Xia J, Zhao C, Hu B, et al. Mole-BERT: Rethinking Pre-training Graph Neural Networks for Molecules[C]//The Eleventh International Conference on Learning Representations.
[2] Zhou G, Gao Z, Ding Q, et al. Uni-Mol: A Universal 3D Molecular Representation Learning Framework[C]//The Eleventh International Conference on Learning Representations.
[3] Xiang H, Zeng L, Hou L, et al. A molecular video-derived foundation model for scientific drug discovery[J]. Nature Communications, 2024, 15(1): 9696.

**Questions:**

1. In Figure 1, molecules with the same structure but different colors are confusing. This should be clarified in the caption.
2. In line 278, the notation "HyG" in z_{j,HyG} appears redundant. Could the authors explain its necessity?
3. How are hyperedge features initialized? If each molecule is represented as a hyperedge, how does the model generalize to unseen molecules during inference?

---

### Official Review · Reviewer_Ciuw · 2025-10-31

**Soundness:** 2
**Presentation:** 3
**Contribution:** 1
**Rating:** 2
**Confidence:** 5

**Summary:**

This paper proposes a new method called FIN-H2AN for representation learning in molecular property prediction tasks. Instead of using molecular graphs or SMILES, the method represents molecules as hypergraphs constructed from three types of binary feature vectors (i.e. molecular fingerprints of MACCS, ErG, and PubChem). Each fingerprint encodes the presence or absence of specific molecular substructure fragments. By designing effective message passing mechanisms over the input hypergraph, the method can efficiently aggregate diverse substructure-level information. Experimental results on MoleculeNet benchmarks demonstrate that FIN-H2AN outperforms typical baseline models.

**Strengths:**

- The paper proposes a method that effectively aggregates three different types of molecular substructure information for representation learning.
- The idea of defining a hypergraph over three fingerprints and designing message passing on that hypergraph is technically very interesting.
- Experiments on the MoleculeNet benchmark show that the proposed method outperforms the baseline models.

**Weaknesses:**

- I don’t think the effectiveness of the proposed method can be fully demonstrated using small-scale benchmarks like MoleculeNet. For example, wouldn’t simply concatenating the three fingerprints and applying a well-tuned XGBoost or LightGBM model achieve similar accuracy? In recent years, it's become widely recognized and discussed [1][2] that for small to medium-sized datasets or typical ADMET classification tasks, fingerprint-based methods combined with tree-based algorithms (like XGBoost, CatBoost, or Random Forest) often outperform more complex GNN models in a more reliable way.
- Some have also pointed out that it's not the method itself that works, but rather the richness of the representation. So if you take multiple fingerprint representations like MACCS, ErG, PubChem, Avalon, and ECFP, and combine them with other molecular descriptor features and embeddings from various GNNs to create a comprehensive feature vector, then properly tune a standard machine learning model (like XGBoost, CatBoost, LightGBM, or Random Forest), the predictive performance might be comparable to the method proposed in this study. Since the proposed method uses three different molecular representations, it's possible that the ensemble effect from this diversity in representation is contributing to its performance. In any case, there are still many open questions and room for discussion, and I don’t think benchmark evaluations at this scale alone are enough to clearly determine the strengths or weaknesses of the proposed method.
- Semantically, each fingerprint is essentially checking for specific substructures. But in constructing the hypergraph, the meaning of each individual bit is ignored, so I don’t think this approach can really be considered as capturing the structure–property relationship of the molecule. It seems more like a mechanical or heuristic way of mixing fingerprint information together.

[1] Could graph neural networks learn better molecular representation for drug discovery? A comparison study of descriptor-based and graph-based models. J Cheminform 13, 12 (2021). https://doi.org/10.1186/s13321-020-00479-8
[2] A systematic study of key elements underlying molecular property prediction. Nat Commun 14, 6395 (2023). https://doi.org/10.1038/s41467-023-41948-6.

**Questions:**

- It’s possible that the input feature vector just happened to work well for the MoleculeNet benchmarks. Did you compare the predictive performance against a simple setup where the three fingerprints are concatenated and a standard machine learning model is tuned using best practices? Or did you test a model that concatenates the three fingerprints with the GNN embeddings you're using for comparison?
- For example, in the OGB Graph Property Prediction (https://ogb.stanford.edu/docs/leader_graphprop/) molecular task *ogbg-molhiv*, the “Molecular FP + Random Forest” baseline actually performs quite well. On that dataset, is your proposed method clearly better than this baseline? And what about using the concatenation of the three fingerprints you used in this study—how does that compare?

---

### Note · Authors · 2026-01-06

I have read and agree with the venue's withdrawal policy on behalf of myself and my co-authors.